# Efficient deep-blue electroluminescence from Ce-based metal halide

Longbo Yang[1,5], Hainan Du[1,5], Jinghui Li[1], Yiqi Luo[1], Xia Lin[1], Jincong Pang[1], Yuxuan Liu[1,2], Liang Gao [1], Siwei He[3], Jae-Wook Kang [3], Wenxi Liang [1,4], Haisheng Song [1], Jiajun Luo [1] ✉ & Jiang Tang [1,4] ✉

Rare earth ions with d-f transitions ($Ce^{3+}$, $Eu^{2+}$) have emerged as promising candidates for electroluminescence applications due to their abundant emission spectra, high light conversion efficiency, and excellent stability. However, directly injecting charge into $4f$ orbitals remains a significant challenge, resulting in unsatisfied external quantum efficiency and high operating voltage in rare earth light-emitting diodes. Herein, we propose a scheme to solve the difficulty by utilizing the energy transfer process. X-ray photoelectron spectroscopy and transient absorption spectra suggest that the $Cs_3CeI_6$ luminescence process is primarily driven by the energy transfer from the $I_2$-based self-trapped exciton to the Ce-based Frenkel exciton. Furthermore, energy transfer efficiency is largely improved by enhancing the spectra overlap between the self-trapped exciton emission and the Ce-based Frenkel exciton excitation. When implemented as an active layer in light-emitting diodes, they show the maximum brightness and external quantum efficiency of 1073 cd m$^{-2}$ and 7.9%, respectively.

Lanthanide ions with parity-allowed d-f transition ($Ce^{3+}$, $Eu^{2+}$) exhibit remarkable optical properties, such as excellent color purity, fast radiation recombination rate, and high light conversion efficiency. However, the localized $4f$ orbitals lead to the low efficiency of carrier injection, and their electroluminescence application is still unsatisfying[1]. The early attempt on rare earth ions adopted alternating current–driven thin film electroluminescence devices, where electrons are accelerated to impact and ionize rare earth ions doped in light-emitting layers[2]. This strategy needs high voltage to accelerate electrons, which leads to a high turn-on voltage (>100 V)[3] and low external quantum efficiencies (<1%)[4,5].

Thus, it is imperative to design suitable energy/charge transfer paths to realize efficient charge injection. Resonance energy transfer is a photophysical process of energy transfer through the dipole-dipole resonance interaction between donors and acceptors, which could be applied to facilitate efficient electrical excitation of the localized $4f$

electron. An ideal energy transfer system needs a close transfer distance and a large spectrum overlap area between the donor and the acceptor. However, in traditional rare-earth doped semiconductor materials such as $LaBr_3:Ce^{3+}$, the low concentration of doped ions makes most randomly created host excitons fall outside of the characteristic Förster dipole-dipole transfer radius, significantly reducing the transfer rate and efficiency[6].

Ce-based metal halide, $Cs_3CeI_6$, is a semiconductor material that satisfies the requirements. Its soft lattice and localized characteristics facilitate the generation of $I_2$-based self-trapped excitons (STEs) similar to the alkali metal halides, while the Ce luminance center behaves as a Ce-based Frenkel exciton (CFE) due to its periodical location in every unit cell[7]. In particular, STE and the surrounding CFE are confined to a $[CeI_6]^{3-}$ octahedron, shortening the energy transfer distance. These properties significantly improve energy transfer efficiency. Moreover, $Cs_3CeI_6$ possesses excellent deep-blue light emission, with

[1]Wuhan National Laboratory for Optoelectronics (WNLO) and School of Optical and Electronic Information, Huazhong University of Science and Technology (HUST), Wuhan, Hubei, China. [2]Hubei Jiufengshan Laboratory, Wuhan, China. [3]Department of Flexible and Printable Electronics, LANL-JBNU Engineering Institute-Korea, Jeonbuk National University, Jeonju, Republic of Korea. [4]Optics Valley Laboratory, Wuhan, Hubei, China. [5]These authors contributed equally: Longbo Yang, Hainan Du. ✉e-mail: luojiajun@hust.edu.cn; jtang@hust.edu.cn

Commission Internationale d'Eclairage (CIE) color coordinates of (0.145, 0.057), close to the Rec. 2020 blue standard (0.131,0.046).

Therefore, $Cs_3CeI_6$-based deep-blue light-emitting diodes (LEDs) deserve further research efforts to realize their full potential[8]. Here, we demonstrate the existence of STE and CFE through XPS studied at the excited states. The energy transfer process between them is characterized by time-resolved photoluminescence (PL) decay and transient absorption. We increase the spectra overlapping area between STE and CFE, accomplished by blue-shifting STE's emission through the addition of excess CsI in the dual-source vacuum thermal co-evaporation process. As a result, the transient absorption spectra exhibit a faster energy transfer (ET) rate (1.4 ps) in line with our expectations. Furthermore, we fabricate $Cs_3CeI_6$-based rare earth LEDs (RELEDs) with improved maximum luminance and external quantum efficiency (EQE) from 530 cd m$^{-2}$ and 3.94% to 1075 cd m$^{-2}$ and 7.9%, respectively, which is the highest efficiency among metal halide deep-blue LEDs so far (Supplementary Table 1). This work fully illustrates the importance of ET engineering in efficient rare earth electroluminescence and provides more opportunities for the next-generation display industry.

## Results

### Energy transfer in $Cs_3CeI_6$

The poor charge injection resulting from $4f$ localization impedes the efficient RELEDs construction. To improve the charge injection process, we first study its basic luminescence mechanism in detail. $Cs_3CeI_6$ shows a cryolite-like structure with the tetragonal space group of P42/m (Fig. 1a). Similar to $[PbI_6]^{4-}$ in cubic phase $CsPbI_3$, $Ce^{3+}$ has a six-coordinated structure and forms a $[CeI_6]^{3-}$ octahedron. The $Cs^+$ ions provide structure support and separate $[CeI_6]^{3-}$ octahedrons to form a zero-dimensional structure. In our previous work, we used the traditional luminescent theory of Ce-doped materials, Ce-$4f$ to $5d$ transition, and the crystal field theory, to explain the luminescence in

$Cs_3CeI_6$. However, this theory still falls short in explaining the electroluminescence process and the excited carrier dynamics. Unlike cerium-doped materials, where isolated Ce ions are luminescent centers, the Ce's periodic occupation in $Cs_3CeI_6$ means Ce-$4f$ orbitals can construct an energy band[9]. The $4f$ energy band makes the Ce-based luminescence behave as excitons rather than isolated ions[7]. Due to the localization of the $4f$ orbital, such excitons usually exhibit the characteristics of Frenkel exciton, such as small excitonic radii and localization at the $Ce^{3+}$, which we here term as CFE. In addition, due to the soft lattice of lanthanide halides, two adjacent $I^-$ in the excited state tend to trap a hole to form an $I_2^-$ $V_k$ center and further trap an electron to form a self-trapped exciton. The standard oxidation-reduction potential (Supplementary Table 2) of $Ce^{3+}$ and $I^-$ also indicates the hole-trapped preference of $I^-$ in $Cs_3CeI_6$. Here, we speculate that the excitation process of $Cs_3CeI_6$ is as follows: $I^-$ sequentially traps holes and electrons to form STE, and then STE excites CFE through the ET process (Fig. 1b). Its representation in the energy band diagram is shown in Fig. 1c.

To confirm the specific luminescence process of $Cs_3CeI_6$, we prepared its crystal using the Bridgman method. The powder XRD results indicate that the crystal phase is pure and consistent with our previous reports (Supplementary Fig. 1)[10]. Since the energy transfer process from STE to CFE is thermally activated, the intrinsic emission of STE stands out when the temperature is lowered. Figure 1d shows the emission spectrum of $Cs_3CeI_6$ at 4.2 K, composed of mixed CFE and STE emissions. The CFE spectrum exhibits two peaks located at 435 nm and 480 nm, respectively, which result from two ground states, $^2F_{5/2}$ and $^2F_{7/2}$, splitting from the spin-orbit coupling of $Ce^{3+}$. The energy difference between the two characteristic emission peaks is 2155 cm$^{-1}$, close to the ground state splitting energy difference of 2000 cm$^{-1}$[11,12]. The luminescence spectrum of STE is obtained by subtracting the CFE spectrum from the total spectrum and ranges from 400 nm to 600 nm. When the temperature returns to room temperature, the STE emission

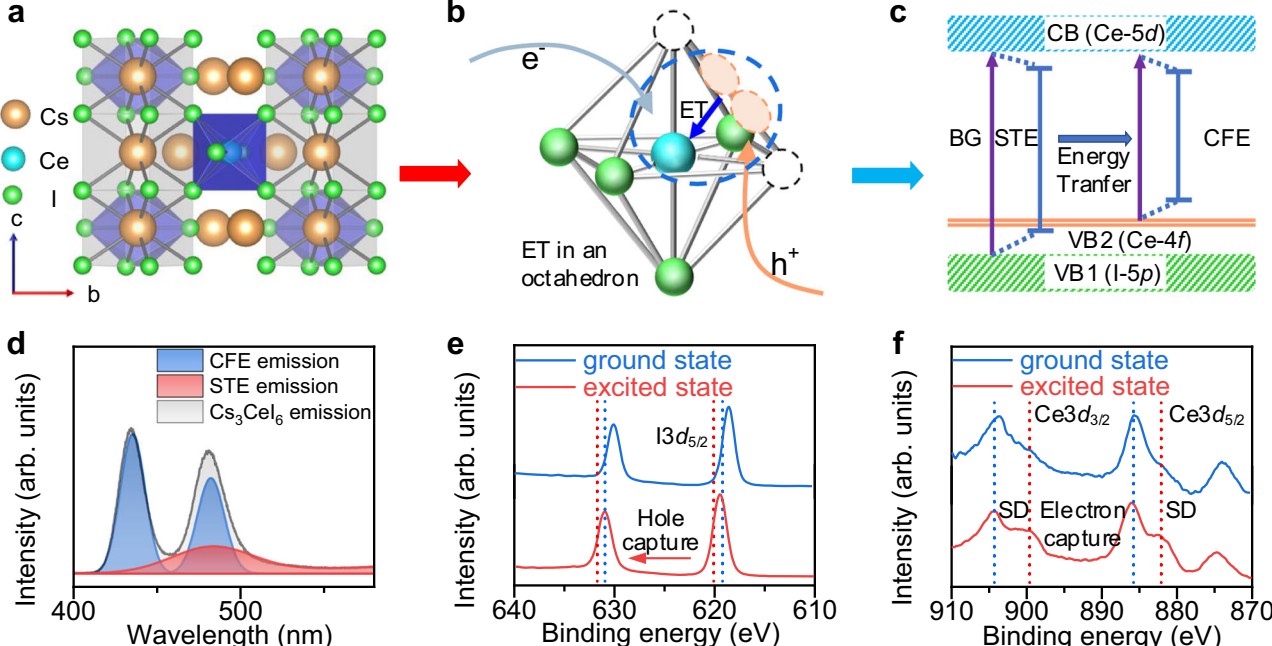

**Fig. 1 | Modeling and characterization of the energy transfer process in $Cs_3CeI_6$.**
**a** Crystal structure of $Cs_3CeI_6$. **b** Schematic diagram of STE formation and energy transfer process. The circular part of the blue dashed line shows the formation of STE and its energy transfer to CFE. ET stands for energy transfer. **c** Schematic diagram of $Cs_3CeI_6$ energy band structure and two types of excitons. The electrons and holes of STE come from CB and VB1 (contributed by Ce-$4f$) respectively, and those of CFE come from CB and VB2 (contributed by I-$5p$) respectively. (BG for

bandgap, STE for self-trapped exciton, CFE for Ce Frenkel exciton, CB for conduction band, and VB for valence band). **d** CFE, STE, and total emission spectra of $Cs_3CeI_6$ crystal in 4.2 K. **e** X-ray photoelectron spectroscopy of I-$3d_{5/2}$ in $Cs_3CeI_6$ in the ground and excited states (in which the excited state is generated by continuous irradiation of 280 nm laser for 5 min). **f** X-ray photoelectron spectroscopy of Ce-$3d_{3/2}$ and Ce-$3d_{5/2}$ in $Cs_3CeI_6$ in the ground and excited states.

gradually weakens because the energy transfer to CFE is thermally activated (Supplementary Fig. 2)[13,14]. In addition, the STE decays rapidly when the temperature is below 100 K, which results from the thermally activated energy transfer of STE to CFE. The intensity of CFE first rises and then falls with the temperature increase because of both energy transfer and temperature quenching. The intensity of CFE1 located at 435 nm and CFE2 located at 480 nm has the same tendency with temperature, which indicates that they originated from a similar recombination channel (Supplementary Fig. 3).

Furthermore, we characterized X-ray photoelectron spectroscopy (XPS) with/without a 280 nm laser irradiation simultaneously onto the sample to study the capture process of photogenerated carriers in the excited state/ground state. It is noted that the photon density of the X-ray during the characterization is much smaller than that of the laser, $Ce^{3+}$ excitation from X-ray irradiation can be ignored. The results show that the XPS peak of $I^-3d_{5/2}$ shifts from 618.58 eV to 619.48 eV, which implies the holes trapping by $I^-$ forming $I_2^- \cdot V_k$ center (XPS peak of $I^-3d_{5/2}$ in CsI is 618.2 eV, and in $I_2$ is 619.9 eV) (Fig. 1e). In comparison, the signal intensity of Ce's shake-down (SD) peak increased relative to the main peak (Fig. 1f). The SD peak signal generally represents the charge transfer state of electrons from ligands to rare earth ions[15]. The enhanced SD signal indicates the tendency to capture electrons. (But it does not mean $Ce^{3+}$ captures electrons to form $Ce^{2+}$). Besides, we also carried out an in-situ laser–off–on–off–on cycle XPS measurement, which showed that the excitation process is reversible (Supplementary Fig. 4).

Based on the above characterization, we have identified STE, CFE, and the ET processes in $Cs_3CeI_6$. Notably, CFE and STE are localized in a $[CeI_6]^{3-}$ octahedron. Thus, the effective transfer distance is estimated at a chemical bond length (~3 Å), much smaller than conventional rare-earth-doped semiconductor materials[16]. However, the emission of STE and the excitation spectrum of CFE overlap very little, which will greatly reduce the energy transfer efficiency between them. Here, we adopt a strategy that adds excessive CsI during $Cs_3CeI_6$ preparation to construct the quantum well structure ($I_2^-$-based STE emission energy is 3.66 eV in CsI[17] and 2.57 eV in $Cs_3CeI_6$, the energy different will promote the confinement affection on STE) (Supplementary Table 3). The STE emission in $Cs_3CeI_6$ is blue-shifted to obtain a higher spectral overlap area through the confinement effect.

## Improving the energy transfer process by excess CsI

We employed dual-source co-evaporation to prepare $Cs_3CeI_6$ films (Fig. 2a). This method can effectively eliminate the interference of solvent pinholes with good stability and repeatability compared with the spin-coating method[18,19]. In addition, this vacuum preparation scheme can effectively avoid $Ce^{3+}$ oxidation. During the dual-source co-evaporation, we set the substrate temperature to 200 degrees to improve the crystallinity of $Cs_3CeI_6$, and the ratio of CsI and $CeI_3$ was precisely controlled by the film thickness meter[20]. Here, we adjusted the ratio of CsI and $CeI_3$ in the evaporation process, slightly increasing it from 3:1 to 3.3:1. The film X-ray diffraction (XRD) spectra show that the film consists of CsI and $Cs_3CeI_6$ without other phases (Supplementary Fig. 5). Energy dispersive spectrometer (EDS) mapping showed that Cs, Ce, and I were distributed evenly on the film. (Supplementary Figs. 6, 7).

We further characterized the optical properties of the control sample ($CsI: CeI_3 = 3:1$) and Cs-rich sample ($CsI: CeI_3 = 3.3:1$). It is noted that although the luminous film has good thermal stability (Supplementary Fig. 8), it is easy to be hydrolyzed in a humid atmosphere (Supplementary Fig. 9). So all the characterization is carried out under the conditions of packaging with quartz sheet. The photoluminescence excitation (PLE) of $Cs_3CeI_6$ can be divided into two bands at 360–420 nm and 200–300 nm (Fig. 2b). The former band exhibits bimodal emission at 373 and 396 nm with an energy difference of 1903 cm$^{-1}$, consistent with the $Ce^{3+}$ two-ground state $^2F_{5/2}$ and $^2F_{7/2}$. So this band belongs to the intrinsic excitation peak of CFE. The latter PLE behaves like band edge excitation, corresponding to the 200–300 nm absorption in Fig. 2c. This segment primarily belongs to STE excitation. Besides, the time-resolved PL curves of the $Cs_3CeI_6$ control sample under these two different excitation bands were also measured (Supplementary Fig. 10). The decay curves under 280 nm excitation demonstrate a slower rising edge than that under 375 nm excitation, which indicates the energy transfer from STE to CFE (Supplementary Fig. 11 and Supplementary Note 1)[21,22].

With the increase of CsI ratio, the CFE excitation peak shifts from 395 to 393 nm with an energy difference of 0.015 eV, and the STE excitation band presents more blueshift from 283 to 275 nm with shift energy of 0.13 eV (Fig. 2b). Correspondingly, the control sample STE emission peak is 483 nm, and that of the CsI-rich sample is 449 nm with

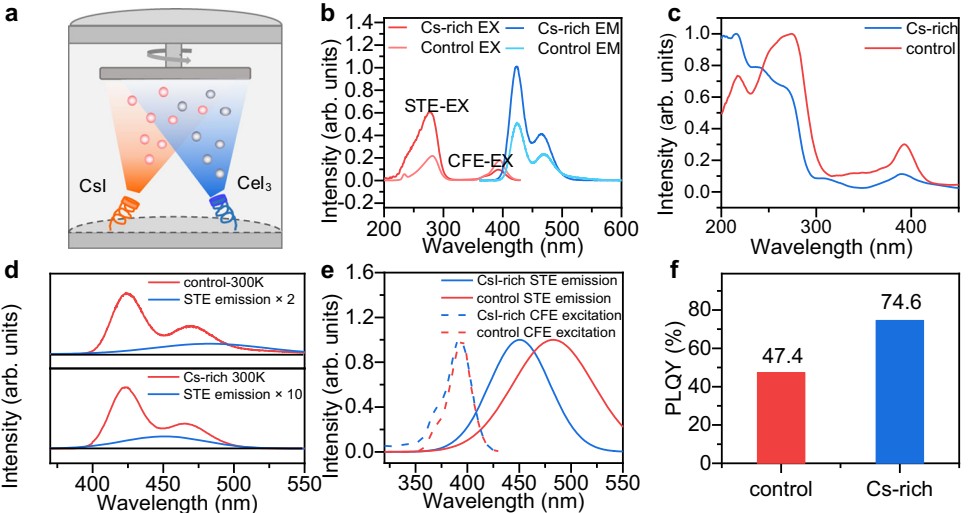

**Fig. 2 | Optical characterization of control and CsI-rich $Cs_3CeI_6$ film. a** Schematic diagram of $Cs_3CeI_6$ dual source co-evaporation. **b** The PL and PLE spectra of the $Cs_3CeI_6$ control and Cs-rich samples (EX for excitation and EM for emission). **c** The absorption spectra of the $Cs_3CeI_6$ control and Cs-rich samples. **d** The PL spectra and the calculated STE emission of the $Cs_3CeI_6$ control and Cs-rich samples. The red line represents the total spectrum and the blue line represents the STE spectrum obtained from the peak-differentiating. (fitted with Gaussian multimodal). **e** The excitation spectra of CFE and the emission spectra of STE and their overlapping areas in $Cs_3CeI_6$ control and Cs-rich samples. **f** The PLQY of 100 nm thick control and Cs-rich film.

energy difference of 0.19 eV, close to their excitation energy difference (Fig. 2d). This blueshift results from the confinement effect on STE in $Cs_3CeI_6$ from the wide-bandgap CsI. To further prove our conclusion, we studied $Cs_3LaI_6$, a material with a similar crystal structure to $Cs_3CeI_6$. Because $Cs_3LaI_6$ demonstrates a $4f$ empty band compared with $Cs_3CeI_6$, its luminescence is a separate STE luminescence. With increasing proportions of CsI, it shows a blueshift tendency consistent with the $Cs_3CeI_6$ (Supplementary Fig. 12)[23].

The blue shift of STE's emission is conducive to the increase of overlap area between STE emission and CFE excitation spectra (Fig. 2e). By integration, we obtained the spectral overlapping areas were $4.56 \times 10^{-30}$ $m^3$ mol for CsI-rich sample and $1.62 \times 10^{-30}$ $m^3$ mol for the control sample, respectively. The effective energy transfer distance varies from 3.18 Å to 3.78 Å as the CsI ratio increases. According to the energy transfer efficiency formula, we can estimate that the energy transfer efficiency increases from 58.8% to 80.6%. (Supplementary Note 2) With efficient ET, the photoluminescence quantum yield (PLQY) of the CsI-rich sample reaches 74.6%, which is higher than that of the control sample at 47.4% (Fig. 2f) (Supplementary Fig. 13). It is worth mentioning here that the CsI-rich PLQY improved by 57% compared to control, but the energy transfer efficiency improved by 37%. In addition to the improvement of energy transfer efficiency, another possible reason for the increase of PLQY is the improvement of film quality. In our previous work, we reported that CsI as a seed layer can improve the crystal quality of thin films[10]. In this work, excessive CsI may also play a significant role in promoting nucleation and crystal growth based on surface atomic force microscopy characterization of the film (Supplementary Fig. 14).

## The dynamics of energy transfer process in $Cs_3CeI_6$

To further explore the dynamics of $Cs_3CeI_6$ exciton at the picosecond scale, we measured the transient absorption spectra (TAS) of films with the $Cs_3CeI_6$ control and Cs-rich samples. It is worth mentioning that selecting a pump laser with energy larger than 4.3 eV is necessary to observe the ET from STE to CFE. Here, we adopt a 250 nm pump laser to observe the dynamic information of the two effective energy bands

described above. The TA UV-visible spectrum at 320–580 nm and the TA visible-infrared spectrum at 550–980 nm are shown in Fig. 3a and b, respectively. Overall, the $Cs_3CeI_6$ transient absorption signals can be divided into three segments. One is the photo-bleach (PB) band from 370 to 395 nm; the second is photo-induced absorption (PIA) signals of the 450–600 nm band; the other is PIA signals of 800–900 nm. The 370–395 nm PB signal corresponding to the PLE band in Fig. 2c comes from the transition from the Ce-$4f$ band to the Ce-$5d$ band, while the I-$5p$ band to the Ce-$5d$ band bleaching signal before 300 nm is out of the measurement range. The PIA signals usually indicate the generation of new energy states, which we believe in the $Cs_3CeI_6$ system results from CFE and STE with different decay rates. Comparing the sequence of PIA signal generation in the second (450–600 nm band) and third segments (800–900 nm), we believe that the second segment is the PIA signal of CFE dissociation, and the third segment comes from the STE distortion (Fig. 3c). In addition, the similar PIA signal of the alkali halides in the infrared band also proves our inference[24,25].

After determining the attribution of TAS, we also measured the TAS of the CsI-rich sample to compare their dynamics processes, as shown in Fig. 3d and e. The results both offer a slow decay in the PB signal, which differs from the rapid attenuation signal under 325 nm excitation in ref. 10 we measured before. The difference comes from the ET from STE to CFE, consistent with the results tested in time-resolved PL decay under different excitation wavelengths (Supplementary Fig. 10). Besides, the rising edge of the CFE PIA signal in the control sample is 1.7 ps, while the STE exciton signal has a rapid decay range from 0.7 ps to 1.7 ps (Supplementary Fig. 15), indicating that the transfer time of STE exciton to Ce exciton is about 1.7 ps (Fig. 3f). This lifetime agrees with the 1 ps transfer rate observed in $CeBr_3$[7]. In the Cs-rich samples, a faster transfer rate of STE to the CFE is observed, and the Ce exciton dissociates earlier (1.4 ps) (Fig. 3f). In addition, the longer lifetime (24 ps) of the $Ce^{3+}$ exciton and a shorter lifetime (6 ps) of the STE indicate that the transfer from STE to Ce exciton is more efficient compared with the control sample (Supplementary Table 4). Those phenomena suggest that the increased overlapping areas improve ET[26]. Moreover, some phenomena in TAS deserve further

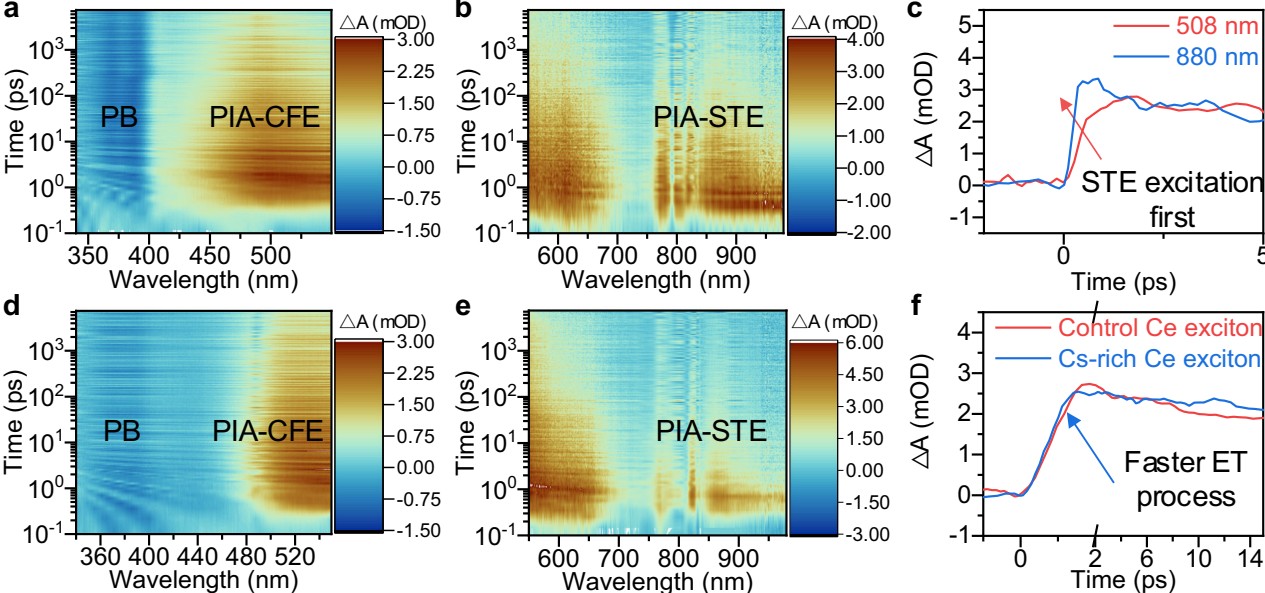

**Fig. 3 | Energy transfer dynamics process in control and CsI-rich $Cs_3CeI_6$ film.** The TAS of the $Cs_3CeI_6$ control sample ranges from 320 nm to 580 nm (**a**) and from 550 nm to 980 nm (**b**). (PB for photo-bleach, PIA for photo-induced absorption, CFE for Ce Frenkel exciton, and STE for self-trapped exciton). **c** The PIA signal in the $Cs_3CeI_6$ control sample at 508 (red line) and 880 nm (blue line). The TAS of the

$Cs_3CeI_6$ Cs-rich sample ranges from 320 nm to 580 nm (**d**) and from 550 nm to 980 nm (**e**). **f** Dynamics processes of CFE in control (red line) and CsI-rich sample (blue line). The pump light is a 250 nm laser with the power of 70 μW when probe light ranges from 320 nm to 580 nm and 290 μW from 550 nm to 980 nm.

study. For example, the PIA signal of CFE may represent the process that the exciton dissociates into an electron and hole then back to the valence and conduction band. The PIA signal of STE is multi-segment, which may be related to the different distortions formed by STE, such as the on-center or off-center $V_k$ center. More details of transient absorption spectra are described in the supplementary information (Supplementary Fig. 16).

### The device performance of $Cs_3CeI_6$-based RELEDs

Thanks to the fast and effective energy transfer process demonstrated above, the difficulty of the carrier's injection into the local Ce-4$f$ orbital is converted into the I-5$p$ orbital, a more delocalized orbit. Therefore, the actual injection bandgap is composed of CB (Ce-5$d$) and VB2 (I-5$p$). The gap value is fitted to 4.2 eV by absorption for the control sample and 4.3 eV for the CsI-rich sample (Supplementary Fig. 17). In addition, since holes are trapped, their transport characteristics are primarily based on the hopping transport mechanism, and the hole mobility (0.202 cm$^2$ V$^{-1}$ s$^{-1}$ obtained in the hole-only device by Space-charge-limited current (SCLC) model) is much lower than the electron mobility (0.905 cm$^2$ V$^{-1}$ ss$^{-1}$ obtained in the electron-only device by SCLC model). (Supplementary Fig. 18) Thus, we need a hole injection layer

with a deeper valence-band maximum (VBM) to enhance the hole injection and balance the hole and electron transport.

Here, we selected the electron transport material ZnO as the electron transport layer and an Al$_2$O$_3$ barrier layer to weaken the electron injection. The N,N-dicarbazolyl-3,5-benzene (mCP) material with a deeper LOMO level (−5.9 eV) was selected as the hole injection layer, a 4,4'-cyclohexylidenebis[N,N-bis(p-tolyl)aniline] (TAPC) layer with high mobility was screened as the hole transport layer, and 4,4',4'-tris(N-carbazolyl)-triphenylamine (TCTA) is sandwiched in the middle to smooth the hole injection (Fig. 4a). The energy level diagram of the device is shown in Fig. 4b, where the Fermi level of $Cs_3CeI_6$ is determined by the UPS and absorption spectrum with the VBM2 at 7.4 eV and CBM at 3.1 eV (Supplementary Figure 19).

We prepared the RELED with the device structure of ITO/ZnO: PEIE (30 nm)/Al$_2$O$_3$ (3 nm)/ $Cs_3CeI_6$ (100 nm)/mCP (10 nm)/TCTA (10 nm)/TAPC (20 nm)/HAT-CN (5 nm)/Al (60 nm). Similar to this structure, we prepared single carrier devices to optimize the injection balance of $Cs_3CeI_6$ RELED. The electron-only device structure is ITO/ZnO: PEIE (30 nm)/Al$_2$O$_3$ (3 nm)/ $Cs_3CeI_6$ /TPBi (5 nm) /LiF (2 nm)/Al (60 nm), and the hole-only device structure is ITO/PEDOT: PSS (20 nm)/ $Cs_3CeI_6$ (100 nm)/mCP (10 nm)/TCTA (10 nm)/TAPC (20 nm)/

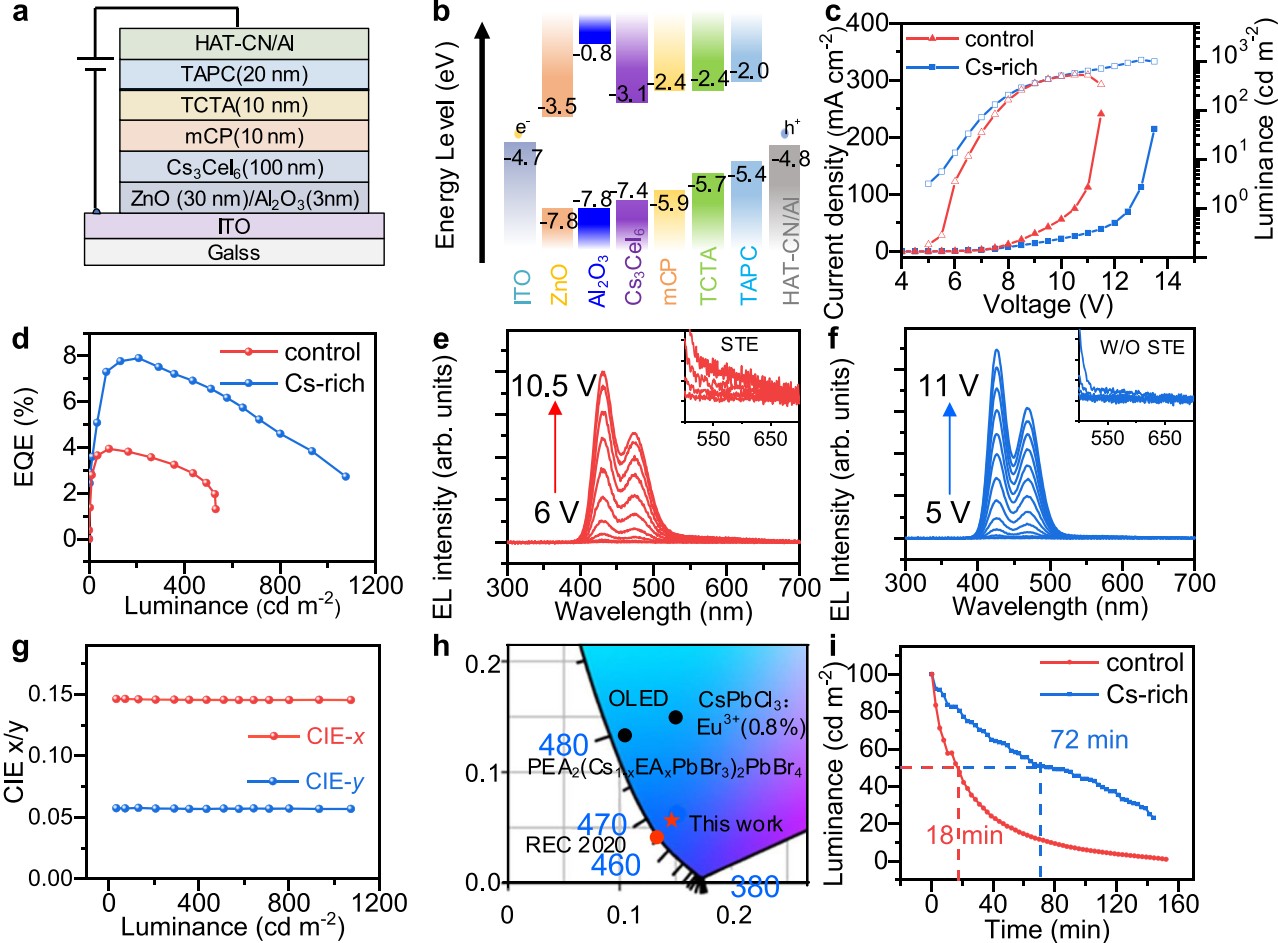

**Fig. 4 | Performance of RELEDs based on control and CsI-rich $Cs_3CeI_6$ films. a** Device structure of $Cs_3CeI_6$ RELED. **b** Flat band energy level diagram of $Cs_3CeI_6$ RELED. **c** Current density-voltage-luminance curve of $Cs_3CeI_6$ control and Cs-rich devices. **d** EQE-luminance curve of $Cs_3CeI_6$ control and Cs-rich devices. **e** EL spectra of the control LEDs under applied voltage from 6 V to 9.5 V. Inset shows the EL spectra of STE after 550−600 nm. The corresponding device voltage and brightness of the EL spectrum taken in the illustration are (7 V, 35 cd m$^{-2}$), (7.5 V, 83 cd m$^{-2}$), (8 V, 164 cd m$^{-2}$), (8.5 V, 260 cd m$^{-2}$). **f** EL spectra of the Cs-rich LEDs under applied voltage from 5 V to 11 V. Inset shows CFE emission spectra without STE after 550 nm.

The corresponding device voltage and brightness of the EL spectrum taken in the illustration are (6.5 V, 33 cd m$^{-2}$), (7 V, 72 cd m$^{-2}$), (7.5 V, 141 cd m$^{-2}$), (8 V, 207 cd m$^{-2}$). **g** The Commission Internationale de l'Éclairage (CIE) coordinates of Cs-rich device's EL spectra at various luminance. **h** The CIE comparison of $Cs_3CeI_6$-RELEDs with Rec. 2020 standard for the blue emitter and some representative deep-blue LEDs reported so far. The CIE values of Ce-complex[11], OLED[31], CsPbCl$_3$:Eu$^{3+}$ nanoparticles[32], and PEA$_2$(Cs$_{1-x}$EA$_x$PbBr$_3$)PbBr$_4$[29] are reported by previous literature. **i** The device stability under a continuous voltage of 8 V for $Cs_3CeI_6$ control devices and 7.5 V for the Cs-rich device.

HAT-CN (5 nm)/Al (60 nm). The results show that the addition of CsI narrowed the gap between the electron and hole current, which may be another reason for better device performance (Supplementary Fig. 20).

The $Cs_3CeI_6$ control device shows a max luminance of 530 cd m$^{-2}$, and it possesses a greater current and lower brightness. In contrast, Cs-rich devices have higher voltage resistance and lower current, with brightness up to 1075 cd m$^{-2}$ (Fig. 4c). The control sample achieved the highest efficiency of 3.94% at 83 cd m$^{-2}$, and the Cs-rich device achieved the highest efficiency of 7.9% at 207 cd m$^{-2}$ (Fig. 4d). That's nearly twice as efficient as the deep-blue Pb-based PeLEDs record and is also much brighter (Table S1)[27,28]. Due to the high turn-on voltage of the device (~5 V), the power efficiency is not ideal, with a maximum of only 1.2 lm W$^{-1}$ for the CsI-rich device and 0.58 lm W$^{-1}$ for the control device (Supplementary Fig. 21). To evaluate the reproducibility of the Cs-rich devices, we count 25 devices with the same preparation technology and obtain an average EQE of 7.52% (Supplementary Fig. 22), manifesting the repeatability of our preparation process. Also, we studied devices with higher CsI components (CsI: $CeI_3$ = 3.5:1), but their spectral overlap area changes very little with poor charge injection. The turn-on voltage increased to 6 V, and the EQE dropped sharply to 3.6% (Supplementary Fig. 23).

As shown in Fig. 4e for the control device, the luminescence of STE (the tail emission from 550 nm to 650 nm) is more evident as the voltage increases, indicating that the transfer ability of STE to Ce exciton is incompetent and cannot satisfy the complete transfer under large charge injection. While in the CsI-rich devices, no distinct STE luminescence was observed (Fig. 4f). Besides, the EL spectrum of the RELED device is very stable and does not change with voltage (Fig. 4g). The CIE coordinate of luminescence is (0.145, 0.057), close to the Rec. 2020 standard of blue (0.131, 0.046) (Fig. 4h). Compared with other Pb-base metal halide perovskites and Ce-complex, it has a significant advantage in CIE coordinate for blue display[11,29]. Further, we measure the stability of $Cs_3CeI_6$ devices, and the $T_{50}$ (operation time for the brightness to decrease to 50% of its initial value) of the control device at the initial luminance of 100 cd m$^{-2}$ was 18 mins, and the $T_{50}$ of Cs-rich device at the initial brightness of 100 cd m$^{-2}$ was 72 mins. (Fig. 4i) The stability of the device is still unsatisfactory, and we believe that the main reason is that the STE formation process traps holes to form $V_k$ centers, which may produce charge accumulations. These charge accumulations may lead to undesirable defects or changes in the valence of $Ce^{3+}$, causing device degradation.

Ce-based metal halide RELEDs with ET engineering solved their charge injection difficulty and initially showed their potential, especially in efficiency optimization and spectrum adjustment (Supplementary Fig. 24). However, many problems have not been fundamentally solved, such as the transport imbalance and energy band mismatch[30]. Nonetheless, considering our limited optimization, we believe the device performance to be encouraging in terms of EQE and stability. More importantly, the ET mechanism extends this electroluminescence strategy to many rare earth halide materials, such as the reported $Cs_3CeBr_6$, $CsEuBr_3$, or some unexplored materials such as zero-dimension Eu-based perovskite $Cs_4EuBr_6$, Ce-based double perovskite $Cs_2NaCeCl_6$, etc.

## Discussion
In summary, we demonstrated efficient deep-blue RELEDs based on Ce-based halide semiconductors, $Cs_3CeI_6$. XPS characterization under excited states confirms that the luminescence is determined by the energy transfer process from STE to CFE. It is worth mentioning that the high Ce atom concentration in the lattice induces its very short transfer distance (~3 Å), which has great advantages over traditional Ce-doped compounds. In this case, we further optimize the spectral overlap area of STE emission and CFE excitation by excessive CsI incorporation, and confirm the increase of energy transfer efficiency by transient absorption and PLQY spectra, respectively. Thanks to the high energy transfer efficiency, $Cs_3CeI_6$ RELED achieves an EQE of 7.9% and a maximum brightness of 1075 cd m$^{-2}$. This performance is encouraging for the RELED based on rare earth inorganic compounds. We believe our work contributes to constructing electroluminescence models for lanthanide halides and can be generalized to other rare earth semiconductor materials.

## Methods
### Materials
Cerium (III) iodide ($CeI_3$, 99.9%) was purchased from Alfa Aesar. Cesium iodide (CsI, 99.999%) and zinc acetate (($Zn(CH_3COO)_2 \cdot 2H_2O$, 99.99%), methoxyethanol, polyethyleneimine-ethoxylated (PEIE, 80% ethoxylated solution, 37 wt% in $H_2O$) were purchased from Sigma–Aldrich. 1,3-Bis(N-carbazolyl)benzene (mCP), 1,1-Bis[4-[N,N'-di(p-tolyl)amino]phenyl]cyclohexane (TAPC), 2,3,6,7,10,11-Hexacyano-1,4,5,8,9,12-hexaazatriphenylene (HAT-CN) were purchased from Xi'an polymer OLED Company. Pattern ITO-coated glass substrates (11 Ω sq$^{-1}$) with a size of 25 mm × 25 mm were purchased from Thin Film Device Inc. All the materials were used as received without further purification.

### Deposition of $Cs_3CeI_6$ film
$Cs_3CeI_6$ film was prepared by the dual-source vacuum thermal co-evaporation in the eight-source TE system (OMVFS300, Fangsheng Optoelectronic Co. Ltd.). CsI and $CeI_3$ are placed in a quartz crucible and vaporized on a diagonal evaporation source. Three crystal oscillators were used to control CsI (1.2 Å s$^{-1}$), $CeI_3$ (0.4 Å s$^{-1}$) evaporation rate, and total film thickness (100 nm) in real-time. At the same time, maintain pressure less than 10$^{-4}$ Pa in the evaporation process, and at the base temperature of 200 °C for in-situ annealing.

### Device fabrication
ITO was cleaned ultrasonically in deionized water, glass detergent, acetone, and ethanol for 30 min successively, and then dried with flowing nitrogen. The ZnO: PEIE precursors were spin-coated at 4000 rpm s$^{-1}$ for 45 s, and annealed at 150 °C for 15 min. Then a 5 nm $Al_2O_3$ film was deposited on the ZnO: PEIE layer by atomic layer deposition (ALD). The ITO/ZnO:PEIE/$Al_2O_3$ substrate is put into the evaporation chamber to evaporate 100 nm $Cs_3CeI_6$ films. After the substrate cools naturally to room temperature, 10 nm mCP, 20 nm TAPC, and 5 nm HAT-CN were sequentially evaporated on it at a 0.2 Å s$^{-1}$ rate. Finally, 60 nm Al was deposited as the pattern anode through the mask. The size of an effective luminous pixel is 2 mm × 2 mm. To ensure the uniformity of the film, the substrate is rotated at 10 rpm throughout the evaporation process. In addition, because $Cs_3CeI_6$ is prone to deliquescence, the evaporation equipment is used in conjunction with the glove box equipment, and LED device testing is conducted in the glove box.

### Material and device characterization
X-ray diffraction experiments were performed by a Philips X'Pert Pro diffractometer with Cu (Kα) radiation ($\lambda$ = 1.54 Å). The steady-state PL, PLE, and time-resolved PL-decay spectra were characterized by the Edinburgh FLS920 system. The absolute PLQY measurements were collected by the quantum yield measurement system with an integrating sphere (Edinburgh FLS980 spectrofluorometer). The absorption spectrum measurement is characterized by an ultraviolet-visible spectrophotometer (SolidSpec-3700).

The ratio of CsI to $CeI_3$ was measured by the ICP-OES (PerkinElmer 8300) by dissolving the film in deionized water. XPS and ultraviolet photoelectron spectroscopy were measured by X-ray Photoelectron spectrometer (AXIS SUPRA+). The surface morphology of the films was characterized by scanning tunneling electron microscopy (ZEISS 5 GeminiSEM 300 field-emission SEM).

Because of the poor absorption of $Cs_3CeI_6$, the films for TA measure were evaporated with a thickness of 1 μm. The TA spectra were measured by the TA spectrometer (Helios, Ultrafast Systems). An amplified Ti: Sapphire (Legend Duo, Coherent Inc.) was used to generate the femtosecond laser pulses of an 800 nm fundamental beam (5-kHz repetition rate and 35 fs pulse width). The 800 nm fundamental beam (5 kHz repetition rate and 35 fs pulse width) is divided into two beams. One forms pump light with 250 nm wavelength through an optical parameter amplifier (TOPASPRIME, Light Conversion), and the other beam forms board UV-visible probe light through $CaF_2$ crystal (ultraviolet band) or visible probe light through the sapphire crystal (visible band). Both the pump and probe pulses were directed into the TA spectrometer.

The *I-V-L* curves, EL spectra, EQE-*L* curves, and EL stability curves are measured by a photoelectric integrated test system (XP-EQEAdv, Xipu Optoelectronics Technology Co., Ltd.), which contains an integrating sphere (Labsphere, GPS-4P-SL), a source meter (Keithley 2400), and a spectrometer (Ocean Optics). All EL tests are performed in the glove box.

## Data availability
The data that support the plots within this paper and other findings of this study are available from the corresponding author upon request.

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

## Acknowledgements

This work was financially supported by the National Natural Science Foundation of China (62250003, 62322505, 62104077, 62104077, 62374069, 62304086), the National Key R&D Program of China (2023YFB3608900, 2021YFB3501800, 2022YFA1204800), Key R&D program of Hubei Province (2021BAA014, 2023BAB102), the Fundamental Research Funds for the Central Universities (2023JYCXJJ040), the Innovation Project of Optics Valley Laboratory (OVL2023ZD002). The authors from HUST acknowledge assistance from the facility support of the Center for Nanoscale Characterization and Devices, the Analytical and Testing Center of HUST, and the Instruments Sharing Platform at the School of Optical and Electronic Information of HUST. We thank the Shanghai Institute of Optics and Fine Mechanics, the Chinese Academy of Sciences for spectroscopic characterization.

## Author contributions

J. T. and J-J. L. supervised the whole project. L-B. Y. and H-N. D. designed and performed most of the experiments, characterizations, and analysis; J-H. L. and Y-Q. L. participated in the EL device fabrication and optimization; X. L. and W-X. L assisted in TA measurements and analysis; J-C. P. carried out the single crystal preparation; Y-X. L. and L. G. were involved in the optical characterizations; S-W. H. and J-W. K. and H-S. S. assisted in device measurement. J-J. L. organized the outline of this paper, and with L-B. Y., H-N. D., and J. T. wrote the paper; All authors discussed the results and commented on the paper.

## Competing interests

The authors declare no competing interests.
