## [Peer Review File · Nature Communications]

Efficient deep-blue electroluminescence from Ce-based metal halideEditorial Note: Parts of this Peer Review File have been redacted as indicated to remove third-party material where no permission to publish could be obtained.

REVIEWER COMMENTS

Reviewer #1 (Remarks to the Author):

Reviewer: In this manuscript, the authors have successfully prepared Cs₃CeI₆ rare earth based materials and demonstrated deep blue LED devices. The Cs₃CeI₆ ReLED achieves an external quantum efficiency of 7.9% and a maximum brightness of 1075 cd/m². This performance is encouraging for the RELED based on rare earth inorganic compounds. I recommend the paper for publication in nature communications after the following points have been addressed:

Comments:

1. This article describes the preparation of Cs₃CeI₆ material using a thermal evaporation method. Can Cs₃CeI₆ material be prepared as a colloidal solution through thermal injection and used to fabricate devices through spin-coating method? Can the author briefly explain the impact of thermal evaporation method and spin-coating method on the preparation of luminescent layer films for devices?
2. The author should also consider the stability of Cs₃CeI₆, such as humidity and thermal stability.
3. The author should provide AFM data for both CsI-rich and control films.

Reviewer #2 (Remarks to the Author):

In this manuscript, the authors have demonstrated efficient deep blue LEDs based Cs₃CeI₆ luminous layer. The LEDs achieve high performance with an external quantum efficiency of 7.9% and a maximum brightness of 1075 cd/m². However, for the blue LEDs assembled from Cs₃CeBr₆ and Cs₃CeBr_{6-x} reported in previous works from the same group (ACS Energy Lett. 2021, 6, 4245–4254; Sci. Adv., 2022, 8, eabq2148), which are similar to the results in this work. The novelty of this work is weak. The reviewer has some comments as follows:

1. The authors previously reported on lighting devices assembled using Cs₃CeBr₆ and Cs₃CeBr_{6-x}, and with Cs₃CeI₆, these three works are very similar. In the previous two works, the authors did not mention STE- and FE-related energy transfer or luminescence; rather, only the d-f transition emission peaks were observed. Why did the author mention FE and STE related luminescence in this work?
2. The different coordination environments of Ce exhibit different rapid d-f transitions related to broad absorption and emission peaks can be adjustable in the regions from the UV to visible and NIR. Therefore, the emission peaks of Ce may be similar to the STE or FE related emission peak. How can the transient absorption signals be distinguished from the exciton signals (FE and STE) rather than from the transitions between different d-f transitions caused by different cerium coordination environments?
3. The luminescence mechanisms of STE and FE were proposed in 2018, clearly describing the theory of the generation of luminescent centers of STE and FE. In this article, there are no clear theoretical results to support the luminescence mechanism. Therefore, the author should provide theoretical results to support the FE related luminescence is different from the d-f transition luminescence.

4. The luminescence of the Ce-ion d-f transition is different from that of other semiconductors. Does Cs₃CeI₆ have similar d-f transition electroluminescent properties? What are the special points to use Cs₃CeI₆ as the luminescence layer? What is the role of Cs in Cs₃CeI₆? Is it feasible to replace Cs with other metal ions?
5. Blue LEDs based on GaN have been developed and commercialized. Compared to commercial blue LEDs, what are the advantages of blue LEDs based on Cs₃CeI₆ in this work?

Reviewer #3 (Remarks to the Author):

The authors demonstrate a rare earth metal halide Cs₃CeI₆-based light-emitting diode, which achieves a high peak external quantum efficiency of 7.9% on deep blue emission, almost the highest-level efficiency of these kinds of devices. The manuscript is quite interesting with encouraging results. The mechanism of emission is studied by decent characterisations. The authors also show how a good device should be designed. I am very excited about the results and support the publication of this ms in Nature Communications, after the authors address the following questions (most of which are minor):

1. Page 5, line 1. The lowest excited state of Cs₃CeI₆ is at around 4.4 eV. Therefore, both 280 nm laser and XPS source would be able to excite the electrons and enable the next process. It is thus unclear that the peak shift of XPS is due to an additional 280 nm laser excitation. It is good to clarify the working powers of the laser and the XPS source. Also, a laser-off-on-off-on cycle XPS measurement in-situ is highly recommended to confirm the effect by the additional laser.
2. Page 5, Fig. 1c. If the STE is contributed by the hole-trapped I⁻, how could VB₂ be due to Ce-4f orbital? Also, it might be good to show both two ground states of CFE in Fig. 1c so that the dual CFE emission peaks can be clearly corresponded.
3. It could be very interesting to show three temperature-dependent photon-integration PL curves, with integrating the STE and the two CFE peaks respectively. Then we would get three values of exciton activated energy, of which the two values from two CFE peaks are supposed to be close, and be different to that of STE. This would be strong evidence that the two major peaks originating from a same recombination channel (CFE) while different from STE channel.
4. Page 8, Fig. 2e. The authors demonstrate an “energy transfer” process from STE emission to CFE excitation. If I understand the mechanisms correctly, in this work, the electrons recombining via STE channel would be at VB₂ and energetically far away from the CFE excited state, only the STE-emissive photons could be re-absorbed via CFE channel. The overlap area in Fig. 2e is in fact the STE-emissive photons being re-absorbed instead of the I⁻ electrons being transferred. The calculation in Text S1 would better be termed as re-absorption efficiency? From Fig. 2b, an energy transfer is hard to occur unless it overcomes the energy gap of ~1.25 eV from STE EX to CFE EX (The authors have shown a correct schematic diagram in Text S2). Maybe the authors can reconsider the description of this process.
5. With the increasing of Cs ratio, the “energy transfer” efficiency increases. Up to Cs ratio of 3.3, there remains a certain gap between STE emission and CFE excitation. Is it possible to continue to increase the ratio of CsI to pursue a higher overlapping of re-absorption?

6. Page 9, line 16. The early research believes that the lifetime of STE is rather long due to the triplet-singlet transition. Can the authors provide more discussions regarding such short lifetimes of I-STE (6 ps) and CFE (24 ps), which is even faster than most singlet-singlet transitions? What would the lifetimes be if the PL decay fitting is used?

7. Page 10, Fig. 3. The signals of PIA-CFE from 3a to 3d shift (or maybe disappear) for almost 100 nm, and the signals of PIA-STE in 3b and 3e are multi-segment. Can authors give any comment on this phenomenon?

8. Page 12, line 14, Fig. 4a. It is very nice that the authors describe how a good device should be designed. For instance, Al₂O₃ barrier is of importance for the sake of lowering electron mobility in this case. I would suggest the authors to label it in Fig. 4a.

9. Luminous power efficiency is worth more attention as it reflects energy efficacy and will be quite important for evaluating the market potential. Can authors provide the data on the luminous power efficiency of Ce₃CeI₃ LED?

Reviewer #1 (Remarks to the Author):

Reviewer: In this manuscript, the authors have successfully prepared Cs₃CeI₆ rare earth based materials and demonstrated deep blue LED devices. The Cs₃CeI₆ RELED achieves an external quantum efficiency of 7.9% and a maximum brightness of 1075 cd/m². This performance is encouraging for the RELED based on rare earth inorganic compounds. I recommend the paper for publication in nature communications after the following points have been addressed.

Response: we appreciate the reviewer for the positive feedback.

Comments:

1. This article describes the preparation of Cs₃CeI₆ material using a thermal evaporation method. Can Cs₃CeI₆ material be prepared as a colloidal solution through thermal injection and used to fabricate devices through spin-coating method? Can the author briefly explain the impact of thermal evaporation method and spin-coating method on the preparation of luminescent layer films for devices?

Response: Thank the reviewer for the professional comment. We believe that Cs₃CeI₆ material can be also prepared using solution methods. Similar material systems have been demonstrated in other reports (*Nano Lett.* 2020, 20, 5, 3734–3739; *J. Phys. Chem. Lett.* 2024, 15, 6, 1668–1676; *Adv. Mater.* 2024, 2310065).

According to our research on the solubility of Cs₃CeI₆, we believe that two problems need to be paid attention to during the preparation of Cs₃CeI₆ colloidal solution. One is that the solvent may compete with I⁻ for the coordination of Ce³⁺, which results in solvent residue in the spin-coated film; the other is the oxidation effect of water and oxygen in the air on Ce³⁺, and it may be necessary to add some reducing agents in the solvent.

The primary advantages of the thermal evaporation method are located in large areas, pixelation, industrial compatibility, and reproducibility. In particular, since most rare earth elements have multiple chemical valences and are easy to coordinate with solvents, the thermal evaporation scheme can effectively avoid these shortcomings, and can ensure a high crystallization film with low defect density. We have now added the corresponding discussions in the revised manuscript.

[Redacted]

Figure 1 Preparation of colloidal solutions by thermal injection in Cs, Eu, and Br systems. (*J. Phys. Chem. Lett.* 2024, 15, 6, 1668–1676)

2. The author should also consider the stability of Cs_3CeI_6 , such as humidity and thermal stability.

Response: Thank the reviewer for the important suggestion. Following the reviewer's suggestion, we have added humidity stability and temperature stability tests' results in Figure S8 and Figure S9.

We note that rare earth halides have strong hygroscopicity, and most can form hydrated halides in the air. Because the humidity stability of Cs_3CeI_6 is poor, the film will dissolve in the air in less than 1 minute. Its thermal stability is very good, and the PL intensity has almost no attenuation after 32 days at 80 °C.

Figure 2 (a, b) The PL spectra of Cs_3CeI_6 film depend on time at 80 °C; (c) The PL intensity of Cs_3CeI_6 film depends on time at 80 °C.

Figure 3 Pictures of Cs₃CeI₆ film: (a) in the glovebox; (b) in the glovebox excited with 365 nm light; (c) in the air for several seconds; (d) in the air for several seconds excited with 365 nm light.

3. The author should provide AFM data for both CsI-rich and control films.

Response: Thank the reviewer for the important suggestion. Following the reviewer's suggestion, we have provided the AFM data for both CsI-rich and control films in Figure S13

Figure 4 The AFM data for CsI-rich (a) and control films (b).

Reviewer #2 (Remarks to the Author):

In this manuscript, the authors have demonstrated efficient deep blue LEDs based Cs₃CeI₆ luminous layer. The LEDs achieve high performance with an external quantum efficiency of 7.9% and a maximum brightness of 1075 cd/m². However, for the blue LEDs assembled from Cs₃CeBr₆ and Cs₃CeBr_xI_{6-x} reported in previous works from the same group (ACS Energy Lett. 2021, 6, 4245–4254; Sci. Adv., 2022, 8, eabq2148), which are similar to the results in this work. The novelty of this work is weak. The reviewer has some comments as follows:

Response: We appreciate the reviewer for the feedback to facilitate the improvement of this manuscript. We understand reviewers' concerns about the innovation of this work, but we note that this work is an extension of previous work rather than a repetitive similar one. In this work, we add a lot of characterization of photophysics in Cs₃CeI₆, such as XPS in excited state and transient absorption, and propose a new energy transfer model from STE to CFE to explain the physical phenomena. The understanding of the photophysical mechanism is very important to guide the optimization of the device, which is also the innovation and importance of this paper compared with previous work. According to the reviewer's suggestion, we also further improved the analysis of transient absorption and the characterization of XPS in this paper. Besides, we provide more explanation of the physical mechanism, so as to improve and highlight the innovation of this work.

1. The authors previously reported on lighting devices assembled using Cs₃CeBr₆ and Cs₃CeBr_xI_{6-x}, and with Cs₃CeI₆, these three works are very similar. In the previous two works, the authors did not mention STE- and FE-related energy transfer or luminescence; rather, only the d-f transition emission peaks were observed. Why did the author mention FE and STE related luminescence in this work?

Response: Thanks for the reviewer's comment. We fully understand the reviewer's concern about the inconsistencies in the interpretation of luminescence between previous and current work. However, we note that the understanding of rare earth luminescence is evolutive due to the complexity of rare earth luminescence. With our in-depth study of rare earth halide luminescence, we found that the previous explanation of Cs₃CeI₆ photophysics was not complete, so we proposed an energy transfer model and provided the evidence with XPS and TA characterization. This model

is also applied in similar compounds, such as CeBr₃ and other scintillators. (*J. Phys.: Condens. Matter* 2006,18,6133; *Phys. Rev. B* 2007 75, 184302; *Phys. Rev. B* 2019, 99, 104301; *Phys. Rev. B* 2018, 97, 144303) We believe that this is a further optimization of the luminescence model without disagreement with previous statements. We have now added the corresponding discussions in the revised manuscript: “*In our previous work, we used the traditional luminescent theory of Ce-doped materials, Ce-4f to 5d transition and the crystal field theory, to explain the luminescence in Cs₃CeI₆. However, this theory still falls short in explaining the electroluminescence process and the excited carrier dynamics process. Thus, we optimized the luminescence model.*”.

[Redacted]

Figure 5 the energy transfer model of LaBr₃: Ce³⁺ in the reference (*Phys. Rev. B* 2007 75, 184302)

2. The different coordination environments of Ce exhibit different rapid d-f transitions related to broad absorption and emission peaks can be adjustable in the regions from the UV to visible and NIR. Therefore, the emission peaks of Ce may be similar to the STE or FE related emission peak. How can the transient absorption signals be distinguished from the exciton signals (FE and STE) rather than from the transitions between different d-f transitions caused by different cerium coordination environments?

Response: Thank the reviewer for the professional comment. The change in Ce coordination environment will indeed bring about the multi-exciton absorption signal, which will be more complex in the emission spectrum. (*J. Mater. Chem. C* 2015, 3 (43), 11366– 11376) However, it should be noted that in the Cs₃CeI₆ crystal with high crystallinity, the Ce coordination environment is consistent, and Ce³⁺ is located in the body center of the [CeI₆]³⁻ octahedron. Thus, it is theoretically improbable to produce multiple Ce coordination environment emissions. In terms of the photophysics of emission peaks, temperature-dependent PL results show that there is a mutual

energy transfer between excitons, and the Stokes displacement of the two excitons is quite different, indicating that the luminescence principle of the two excitons is highly inconsistent. In addition, in terms of the characterization of transient absorption, the PIA signal of I_2^- is similar to that reported in previous literature, and the energy location is the same, which is a direct evidence to prove the existence of STE. We now provide more discussions of transient absorption in the revised manuscript.

Figure 6 Temperature-dependent photon-integration PL curves with integrating the STE and the two CFE peaks respectively. CFE1 and CFE2 correspond to the emission at 435 nm and 480nm, respectively.

[Redacted]

Table 1 Peak positions of the electronic transition bands in the self-trapped excitons (Ref: Phys. Rev. B 1993, 47, 6747)

3. The luminescence mechanisms of STE and FE were proposed in 2018, clearly describing the theory of the generation of luminescent centers of STE and FE. In this article, there are no clear theoretical results to support the luminescence mechanism. Therefore, the author should provide theoretical results to support the FE related luminescence is different from the d-f transition luminescence.

Response: Thank the reviewer for the professional comment. We have provided the band structure

in the ground state in our previous work (*Science Advances*. 2022, 8, q2148), but it is difficult to calculate excited states because it involves the formation of two coupled strongly confined excitons (CFE and STE) and the distortion of the lattice. Some reports have indirectly estimated the formation tendency of self-trapped excitons by calculating the effective mass. (*Journal of Luminescence*. 2021, 237, 118147; *Materials*. 2021, 14, 4243) Some theoretical calculations attribute the difficulties in STE to the lack of “*advanced many-body theory and lattice relaxation mechanism*”. (*Physical Review B*, 2011, 83, 125115) Therefore, most researches focus on experimental results to support STE's conclusions. We have provided the existence of STE from the experimental results by the temperature-dependent photon-integration PL curves in Figure 6. Besides, we believe XPS data is the direct evidence and explain it in the manuscript: “*The results show that the XPS peak of $I-3d_{5/2}$ shifts from 618.58 eV to 619.48 eV, which implies the holes trapping by I forming $I_2^- V_k$ center*”. We agree with the reviewer's need for theory, and hope that there will be more theoretical studies in the future to verify the model proposed in our work.

[Redacted]

Figure 7 (a) Calculated electronic band structure of Cs_3CeI_6 . (b) Schematic diagram of Cs_3CeI_6 Ce-5d \rightarrow Ce-4f parity-allowed emission. (Ref: *Science Advances*. 2022, 8, q2148)

4. The luminescence of the Ce-ion d-f transition is different from that of other semiconductors. Does CeI_3 have similar d-f transition electroluminescent properties? What are the special points to use Cs_3CeI_6 as the luminescence layer? What is the role of Cs in Cs_3CeI_6 ? Is it feasible to replace Cs with other metal ions?

Response: Thank the reviewer for the professional comment. Generally, the emission of 4f-5d in CeI_3 is very weak (*Journal of Crystal Growth* 2020, 531, 125365). Because “*In the case of CeI_3 the 5d1 and 5d2 subbands are overlapped, that makes 5d \rightarrow 4f luminescence observation complicated at room temperature.*” (*Journal of Luminescence* 2021, 237, 118147) The role of Cs is partly to

provide structural support, and on the other hand also to change the energy gap of the semiconductor. Moreover, an efficient energy transfer process can be observed in Cs_3CeI_6 with a suitable band structure. Cs can be substituted, for example Rb_3CeI_6 can also be used as an emissive layer (Opt. Lett. 2023, 48, 2777-2780).

[Redacted]

Figure 8 Optical properties of the Rb_3CeI_6 film. (Ref: Opt. Lett. 2023, 48, 2777-2780)

5. Blue LEDs based on GaN have been developed and commercialized. Compared to commercial blue LEDs, what are the advantages of blue LEDs based on Cs_3CeI_6 in this work?

Response: GaN LEDs are usually used in the lighting field and some large screen display fields. GaN films are prepared on sapphire substrate by MOCVD at in-situ high temperatures, so they cannot be directly deposited onto CMOS or TFT substrates, which need to use costly LED massive transfer technology for display application.

The advantages of rare earth LED are mainly reflected in three aspects. One is the scientific value: Research on rare earth materials charge injection is conducive to deepening the understanding of 4f electrons. The second is the application value: the rare earth halide can be prepared by thermal evaporation which is compatible with the existing OLED industry (Nature Photonics 2023, 17,435–441). The other is the expansion potential, the low-temperature integration process of Ce-based halide materials makes them have potential in the application of sensor light source and on-chip light source.

Reviewer #3 (Remarks to the Author):

The authors demonstrate a rare earth metal halide Cs₃CeI₆-based light-emitting diode, which achieves a high peak external quantum efficiency of 7.9% on deep blue emission, almost the highest-level efficiency of these kinds of devices. The manuscript is quite interesting with encouraging results. The mechanism of emission is studied by decent characterizations. The authors also show how a good device should be designed. I am very excited about the results and support the publication of this ms in Nature Communications, after the authors address the following questions (most of which are minor):

Response: We appreciate the reviewer for the positive feedback.

1. Page 5, line 1. The lowest excited state of Ce³⁺ is at around 4.4 eV. Therefore, both 280 nm laser and XPS source would be able to excite the electrons and enable the next process. It is thus unclear that the peak shift of XPS is due to an additional 280 nm laser excitation. It is good to clarify the working powers of the laser and the XPS source. Also, a laser-off-on-off-on cycle XPS measurement in-situ is highly recommended to confirm the effect by the additional laser.

Response: Thank the reviewer for the professional comment. Following the reviewer's suggestion, we have provided the laser-off-on-off-on cycle XPS measurement in-situ measure in supplementary information. The results show that it is completely reversible and not affected by X-ray excitation in the XPS results. We believe that this possible reason is that the photon density of the X-ray (power 70 W~600 W, photon energy <6 keV) during the characterization is much smaller than that of the laser (power 40 W, photon energy 4.42 eV) which leads to the XPS peak movement of Ce mainly from ultraviolet excitation. In addition, some literature has reported the XPS stability of Ce³⁺, which also indicates that Ce³⁺ XPS is generally not affected during testing. (*Nanoscale* 2012, 4, 16, 4950-4953)

Figure 9 (a) laser-off-on-off-on cycle XPS measurement in-situ of Ce-3d_{3/2} and Ce-3d_{5/2}. (b) laser-off-on-off-on cycle XPS measurement in-situ of I-3d_{5/2}. Stage 1,2,3,4 stands for ground/excited/ground/excited states. The adjacent stages are tested five minutes apart.

2. Page 5, Fig. 1c. If the STE is contributed by the hole-trapped I⁻, how could VB2 be due to Ce-4f orbital? Also, it might be good to show both two ground states of CFE in Fig. 1c so that the dual CFE emission peaks can be clearly corresponded.

Response: Thank the reviewer for the professional comment. According to the reviewer's suggestion, we modified Figure 1c to express the double ground state of Ce more intuitively. We find that Figure 1c is easy to mislead VB2 as a distorted state of STE, so an explanation is added to the figure note, emphasizing that VB2 corresponds to the ground state band of CFE.

Figure 10 Schematic diagram of Cs₃CeI₆ energy band structure and two types of excitons. The electrons and holes of STE come from CB and VB1 respectively, and those of CFE come from CB and VB2 respectively. (BG for band gap, STE for self-trapped exciton, CFE for Ce Frenkel exciton)

3. It could be very interesting to show three temperature-dependent photon-integration PL curves, with integrating the STE and the two CFE peaks respectively. Then we would get three values of

exciton activated energy, of which the two values from two CFE peaks are supposed to be close, and be different to that of STE. This would be strong evidence that the two major peaks originating from a same recombination channel (CFE) while different from STE channel.

Response: Thank the reviewer for the professional comment. According to the review's comment, we calculate the three temperature-dependent photon-integration PL curves in Figure S12. The similar temperature-dependent intensity curves of CFE bimodal indicate that the two major peaks originate from the same recombination channel. It is worth mentioning that due to the influence of the energy transfer process between STE and CFE, we believe that the curve of its intensity with temperature should be determined by the energy transfer and the temperature quenching. For STE, the activation energy is 7.84 meV, and we believe that the activation energy is primarily determined by the energy transfer process. For CFE1 and CFE2, we fit their activation energies at high temperatures. Under this condition, the energy transfer process has been completely activated without the effect of temperature. The activation energies of CFE1 and CFE2 are 192 meV and 228 meV, respectively. We believe that this part is mainly caused by temperature quenching. The small difference in activation energy may result from the different positions of the ground state relative to the valence band.

Figure S3 (a) Temperature-dependent photon-integration PL curves with integrating the STE and the two CFE peaks in crystal, respectively. CFE1 and CFE2 correspond to the emission at 435 nm

and 480 nm, respectively. (b) Fitting diagram of STE activation energy. (c) Fitting diagram of CFE1 activation energy ($T > 120$ K). (d) Fitting diagram of CFE2 activation energy. ($T > 120$ K).

4. Page 8, Fig, 2e. The authors demonstrate an “energy transfer” process from STE emission to CFE excitation. If I understand the mechanisms correctly, in this work, the electrons recombining via STE channel would be at VB2 and energetically far away from the CFE excited state, only the STE-emissive photons could be re-absorbed via CFE channel. The overlap area in Fig. 2e is in fact the STE-emissive photons being re-absorbed instead of the I-electrons being transferred. The calculation in Text S1 would better be termed as re-absorption efficiency? From Fig. 2b, an energy transfer is hard to occur unless it overcomes the energy gap of ~ 1.25 eV from STE EX to CFE EX (The authors have shown a correct schematic diagram in Text S2). Maybe the authors can reconsider the description of this process.

Response: Thank the reviewer for the professional comment. We appreciate the reviewer's suggestions on the accuracy of our physical process description. We added this part of the reabsorption description to textS1. However, we emphasize that reabsorption is also an energy transfer process, primarily occurring during the Forster energy transfer process. In general, the energy transfer process can occur not only through reabsorption but also through carrier transfer, which mainly occurs in the Dexter transfer process. The dipole-dipole Forster energy transfer process described in our manuscript is a reabsorption process from the spectral point of view, but at the molecular level, it occurs in the form of dipole-dipole resonance of excitons, which is two expressions of a transfer process and does not conflict.

5. With the increasing of Cs ratio, the “energy transfer” efficiency increases. Up to Cs ratio of 3.3, there remains a certain gap between STE emission and CFE excitation. Is it possible to continue to increase the ratio of CsI to pursue a higher overlapping of re-absorption?

Response: Thank the reviewer for the professional comment. When the proportion of CsI increases, the spectral overlap area changes little, but the conductivity of the film decreases seriously. The turn-on voltage is up to 6 V, and the device performance decreases accordingly. We add a description on the energy transfer efficiency and conductivity of the film as the CsI: CeI₃ ratio changes.

Figure 12 PL spectra with different Cs: Ce ratio (a) and amplification at STE emission band (b).

(c) Current density-voltage-luminance curve with Cs: Ce=3.5:1.

6. Page 9, line 16. Early research believes that the lifetime of STE is rather long due to the triplet-singlet transition. Can the authors provide more discussions regarding such short lifetimes of I-STE (6 ps) and CFE (24 ps), which is even faster than most singlet-singlet transitions? What would the lifetimes be if the PL decay fitting is used?

Response: Thank the reviewer for the professional comment. Dipole-dipole resonance energy transfer process is a fast process, and the time of its occurrence is usually in the order of ps. (*J. Am. Chem. Soc.* 2021, 143, 11, 4244–4252). Therefore, transient absorption observation is used in this work. The luminescence process involves many factors such as exciton composition, phonon participation, transition mode, etc. The lifetime is usually in the order of ns. The two processes are usually not comparable. The process of energy transfer of ps magnitude cannot be observed in PL decay, but the result of energy transfer can be observed.

7. Page 10, Fig. 3. The signals of PIA-CFE from 3a to 3d shift (or maybe disappear) for almost 100 nm, and the signals of PIA-STE in 3b and 3e are multi-segment. Can authors give any comment on this phenomenon?

Response: Thank the reviewer for the professional comment. There are many theories about the origin of the photo-induced absorption signal of the TAS, including hot carrier relaxation and excited state recombination. Here we attribute the absorption signal to the exciton dissociation to the valence band, which is caused by the distorted recombination of the excited state of the Frenkel exciton. Therefore, this signal movement may result from a change in dissociation energy. The emission spectrum of CsI-rich sample is blue-shifted resulting the lower dissociation energy and the redshift

of the PIA signal. In addition, the multi-stage peaks of Figure 3d and Figure 3e can result from multiple types of distortion in STE, such as on-center and off-center. (Phys. Rev. B 2019, 99, 104301; Phys. Rev. B 2018, 97, 144303)

8. Page 12, line 14, Fig. 4a. It is very nice that the authors describe how a good device should be designed. For instance, the Al_2O_3 barrier is of importance for the sake of lowering electron mobility in this case. I would suggest the authors to label it in Fig. 4a.

Response: Thank the reviewer for the professional comment. According to the reviewer's suggestion, we have modified Fig. 4a in the revised manuscript.

Figure 13 (a) Device structure of Cs_3CeI_6 RELED. (b) Flat band energy level diagram of Cs_3CeI_6 RELED.

9. Luminous power efficiency is worth more attention as it reflects energy efficacy and will be quite important for evaluating the market potential. Can authors provide the data on the luminous power efficiency of Ce_3CeI_6 LED?

Response: Thank the reviewer for the professional comment. According to the reviewer's suggestion, we added the energy efficiency of the luminescence.

Figure 14 Powder Efficiency-voltage curve of Cs_3CeI_6 control and Cs-rich devices.

REVIEWERS' COMMENTS

Reviewer #1 (Remarks to the Author):

The authors have answered my questions. I recommend to accept this manuscript.

Reviewer #2 (Remarks to the Author):

The reviewer's concern is still about the novelty and significance, since the similar concept has been well demonstrated in previous articles from same group, such as ACS Energy Lett. 2021, 6, 4245–4254; Sci. Adv., 2022, 8, eabq2148.

Reviewer #3 (Remarks to the Author):

The authors have addressed all my concerns and I hence recommend to publish it as is.

Reviewer #2 (Remarks to the Author):

Comment 2:

The reviewer's concern is still about the novelty and significance, since the similar concept has been well demonstrated in previous articles from same group, such as ACS Energy Lett. 2021, 6, 4245–4254; Sci. Adv., 2022, 8, eabq2148.

Response: We appreciate the reviewer for the feedback to facilitate the improvement of this manuscript. We understand reviewers' concerns about the innovation of this work, but we note that this work is an extension of previous work rather than a repetitive similar one. Compared with the previous preliminary exploration of Cs₃CeBr₆ (ACS Energy Lett. 2021, 6, 4245–4254) and Cs₃CeBr_xI_{6-x} (Sci. Adv., 2022, 8, eabq2148) material systems, this work focuses on the luminescence mechanism of Cs₃CeI₆, and characterized the carrier dynamics process of their excited states. The transient absorption and excited state of XPS provide strong evidence for the explanation of photophysics. Moreover, the maximum EQE of Cs₃CeI₆-LEDs have been improved to 7.9%, more than twice as the efficiency in previously reported (Sci. Adv., 2022, 8, eabq2148), which is a record value for lead-free blue PeLEDs. According to the reviewer's suggestion, we emphasize the differences from the previous articles in discussion and provide more explanation of the physical mechanism, so as to improve and highlight the innovation of this work.